# The Right Ventricle in COVID-19

**DOI:** 10.3390/jcm10122535

**Published:** 2021-06-08

**Authors:** Jean Bonnemain, Zied Ltaief, Lucas Liaudet

**Affiliations:** 1Department of Adult Intensive Care Medicine, Lausanne University Hospital, 1011 Lausanne, Switzerland; jean.bonnemain@chuv.ch (J.B.); zied.ltaief@chuv.ch (Z.L.); 2Division of Pathophysiology, Faculty of Biology and Medicine, University of Lausanne, 1011 Lausanne, Switzerland

**Keywords:** COVID-19, ARDS, right ventricle, pulmonary circulation

## Abstract

Infection with the novel severe acute respiratory coronavirus-2 (SARS-CoV2) results in COVID-19, a disease primarily affecting the respiratory system to provoke a spectrum of clinical manifestations, the most severe being acute respiratory distress syndrome (ARDS). A significant proportion of COVID-19 patients also develop various cardiac complications, among which dysfunction of the right ventricle (RV) appears particularly common, especially in severe forms of the disease, and which is associated with a dismal prognosis. Echocardiographic studies indeed reveal right ventricular dysfunction in up to 40% of patients, a proportion even greater when the RV is explored with strain imaging echocardiography. The pathophysiological mechanisms of RV dysfunction in COVID-19 include processes increasing the pulmonary vascular hydraulic load and others reducing RV contractility, which precipitate the acute uncoupling of the RV with the pulmonary circulation. Understanding these mechanisms provides the fundamental basis for the adequate therapeutic management of RV dysfunction, which incorporates protective mechanical ventilation, the prevention and treatment of pulmonary vasoconstriction and thrombotic complications, as well as the appropriate management of RV preload and contractility. This comprehensive review provides a detailed update of the evidence of RV dysfunction in COVID-19, its pathophysiological mechanisms, and its therapy.

## 1. Introduction

Since the first cases of infection with the novel severe acute respiratory coronavirus-2 (SARS-CoV2) in December 2019, more than 160 million cases of COVID-19 and 3.3 million deaths worldwide have been reported (as of 12 May 2021). While the primary clinical manifestations of COVID-19 involve the respiratory tract, it has now become evident that SARS-CoV2 provokes a systemic disease with multiorgan involvement. Among the extrapulmonary manifestations of COVID-19, a number of cardiac complications have been described, including myocarditis, taksotsubo cardiomyopathy, arrhythmias, or acute coronary syndromes. In addition, dysfunction of the right ventricle (RV) has emerged as a common feature of COVID-19, most significantly in patients with acute respiratory distress syndrome (ARDS), responsible for the majority of casualties in COVID-19. Although RV dysfunction is a known complication of ARDS from various etiologies, it seems particularly prevalent in the context of COVID-19. Several pathophysiological processes set in motion during the infection with SARS-CoV2 permit us, at least partly, to explain the frequent occurrence of RV dysfunction in this condition. The aim of the present comprehensive review is to provide an update of the relevant literature pertaining to the clinical presentation, the pathophysiology, and the therapeutic management of RV dysfunction in COVID-19.

## 2. Methods

This article is a comprehensive review of the relevant literature pertaining to RV dysfunction in the setting of COVID-19. We included articles published in the PubMed database in 2020 and 2021 using the following search strategies: “COVID 19 and Echocardiography”, “COVID-19 and right ventricle”, “COVID-19 and Pulmonary Circulation”, “COVID-19 and Pulmonary Hypertension”. A total of 886 articles were retrieved, which were selected after reading the abstracts for the relevance of their content with respect to the current review. We also screened the bibliography from selected articles to obtain additional references not obtained from the initial search. We finally included 151 articles for the review process.

## 3. Right Ventricular Dysfunction in COVID-19: What Is the Evidence?

Creel-Bulos et al. reported the first series (21 May 2020) of acute RV failure in five COVID-19 patients hospitalized for respiratory failure management in the ICU. All five patients developed acute hemodynamic instability, with cardiac arrest due to pulseless electrical activity in four patients, 2–9 days after their ICU admission. All displayed severe RV dilation with paradoxical septal movement and RV systolic dysfunction (acute cor pulmonale) due to suspected pulmonary embolism in each case. Systemic thrombolysis was administered to two patients, who survived the acute episode, whereas all three patients not treated with thrombolytics died [1].

These observations were concomitant to several reports from China and Italy reporting a high incidence of acute RV dysfunction in severe COVID-19 patients. Rauch et al. published a series of 29 ICU patients hospitalized from March to April 2020 [2], reporting five cases with elevated troponin (TnI) and echocardiographic features consistent with myopericarditis, as shown by pericardial effusion around the right cavities and fibrinous exudates within the right-sided segment of the atrio-ventricular sulcus. Three out of the five cases also displayed marked RV dilation, although without signs of RV pressure overload and without pulmonary hypertension, supporting myocarditis as the causal mechanism of the RV involvement in this series. In a study on 332 COVID-19 patients hospitalized from February to April 2020, Ferrante reported an incidence of 37% of myocardial injury, as defined by TnI > 20 ng/L, that was independently associated with an increased risk of death (adjusted OR 2.25, *p* < 0.005). In a subset of 21 patients, echocardiography displayed RV dilation as the primary alteration in patients with elevated TnI. In addition, a greater diameter of the pulmonary artery on CT scan, with a cut-off value of 32 mm, was independently associated with myocardial injury and death. Overall, these findings suggested that ongoing damage to the pulmonary circulation might be a critical trigger of myocardial injury and RV dysfunction in COVID-19 [3].

Soon after these observations, a series of more detailed echocardiographic investigations in COVID-19 patients were released, summarized in Table 1. Usual criteria used for the diagnosis of RV dilation and dysfunction in these studies included: RV/LV end diastolic area > 0.6 (>1 for severe dilation); RV diameter > 42 mm (RV base). TAPSE ≤ 17 mm (tricuspid annular plane systolic excursion); S′ < 10 cm/s (tissue doppler-derived tricuspid lateral annular systolic velocity); RV FAC < 35% (RV fractional area change), and RV EF < 45% (RV ejection fraction) [4].

Argulian et al. reported their initial experience with echocardiographic monitoring of 110 COVID-19 patients hospitalized at Mount Sinai Morningside Hospital in NYC in Spring 2020, including 31 intubated patients in the ICU. The most remarkable finding was RV dilation in 32 patients (31%) with RV hypokinesis in about 2/3, while LV function was not affected. RV dilation was the only variable associated with mortality in multivariable analysis, with 41% vs. 11% deaths in patients with or without RV dilation, respectively [7]. Also, in 66 patients explored by echocardiography for cardiorespiratory failure or shock, including a majority of ICU patients, Schott et al. reported that the main echocardiographic abnormality was RV dilation in 74%, while only a minority of patients displayed signs of impaired LV function [6]. Further evidence of RV involvement in COVID-19 was obtained by D’Andrea et al., who reported a significant association between RV dilation/systolic dysfunction and mortality (50% vs. 8%) in a cohort of 115 patients, including 26% under mechanical ventilation. Patients with RV dysfunction had signs of myocardial injury (elevated TnI), more severe lung disease, and higher pulmonary artery pressure, suggesting increased afterload as the primary mechanism of RV dysfunction in these patients [13]. In line with these findings, Pagnesi et al. reported a high prevalence (12%) of pulmonary arterial hypertension in a cohort of 200 non-ICU COVID-19 patients, as well as a 14.5% prevalence of RV systolic dysfunction. Both were associated with cardiac co-morbidities and higher plasma levels of cardiac biomarkers (TnI and NT-proBNP), but only PH was associated with more severe lung disease. Interestingly, PH, but not RV dysfunction, was significantly associated with mortality (42% vs. 9%) in this cohort of non-critically ill patients [14].

Szekely et al. presented a very detailed prospective echocardiographic study in 100 patients hospitalized in Tel-Aviv in March and April 2020 [9]. Echocardiography performed within the first 24 h was abnormal in as much as 68% of patients, with RV dilation and systolic dysfunction being the most frequent alteration (39%), as assessed by TAPSE, RVS′, and RV fractional area change [29]. In contrast, LV systolic and diastolic dysfunction were noted in only 10% and 16% of patients, respectively. RV dysfunction was associated with more severe lung disease and worse prognosis, higher TnI levels, and a reduced pulmonary artery acceleration time, indicative of increased pulmonary vascular resistance. Clinical deterioration was accompanied by further RV dysfunction and was associated with suspected PE in about half of deteriorating patients [9]. Soon after this publication, Kim et al. released data from 510 patients, including 68% ICU patients, undergoing echocardiography in three NY hospitals from March to May 2020 [20]. RV dilation was the most frequent finding, present in 35% of the cohort, while RV systolic dysfunction was found in 15%, and these changes were associated with higher levels of circulating biomarkers (TnI and D-dimers). RV abnormalities were associated with lower LVEF, were more common in mechanically ventilated patients, and were associated with higher systolic pulmonary artery (PA) pressure, supporting increased pulmonary vascular load as the primary pathophysiological mechanism. In multivariate analysis, RV abnormalities independently predicted mortality (32% of the cohort).

More recently, Bleakley et al. reported crucial novel information on a peculiar pattern of RV dysfunction in COVID-19 [21]. In a study exclusively performed in 90 critically ill, mechanically ventilated patients, including 42% on VV-ECMO, these authors found a specific phenotype of RV *radial* dysfunction in more than 70% of patients, characterized by very significant reduction of RV FAC and RV velocity time integral (RV VTI). This contrasted with indices of *longitudinal* dysfunction, as evaluated by RVS′ and TAPSE, and RV free wall strain, which were abnormal in only 24%, 12%, and 35%, respectively. Furthermore, using an indirect index of RV-PA coupling (see below), calculated as the ratio of RV FAC/RV systolic pressure, they found RV/PA uncoupling (defined by a ratio < 1) in 86% of patients, qualified as severe (<0.6) in 47% of patients. These findings indicate that most patients with severe COVID-19 display evidence of RV dysfunction and RV-PA uncoupling, which can be easily overlooked if only usual indices of longitudinal RV function (TAPSE, S′) are taken into account, as this specific form or RV dysfunction seems to spare longitudinal function.

It is particularly noteworthy that the usual parameters used to assess RV function may be insensitive to subtle changes of RV function, as they only represent a small portion of the RV ventricle, address only RV contraction in its longitudinal component, and are dependent to the angle of their measurements. Newly developed indices of ventricular function, based on two-dimensional speckle tracking echocardiography (2D-STE) allow for description of myocardial strain (deformation), which is an important marker of early and subclinical ventricular systolic dysfunction [30]. Notably, RV strain (evaluated by RV longitudinal strain, RV LS, and free wall longitudinal strain, RV FWLS) holds prognostic value in heart failure [31], pulmonary hypertension, and ARDS [32]. In the latter, RV strain has been shown to provide significant diagnostic values when compared to more conventional indices of RV function [33].

The role of strain analysis to detect RV dysfunction in COVID-19 has been the matter of several recent investigations summarized in Table 2, as well as a systematic review and meta-analysis [30]. The most detailed studies have included 462 patients overall. In the first one, Li et al. showed that RV LS detected RV dysfunction more sensitively and accurately than TAPSE and RV FAC, was more severely reduced in mechanically ventilated patients exhibiting higher PA pressure, and that a value of RV LS < −23% was highly and significantly associated with mortality [34]. In the second one, 214 non-CU patients were compared with matched control in a large prospective study in Denmark (ECHOVID study). Most COVID patients displayed signs of reduced LV and RV function, assessed by global strain, RV LS and TAPSE, and these indices were independently associated with mortality [35]. In the third one, Zhang et al. combined 2D-STE strain evaluation with the determination of 3D-RV ejection fraction, which comprehensively evaluates all parts of the RV, including inflow, apical, and outflow portions, in 128 ICU patients. These two indices were both independent predictors of mortality, with 3D-RVEF being the most robust, indicating that this parameter could provide incremental prognostic value over conventional echocardiography in COVID-19 patients [36].

It is important to emphasize that some of the echographic abnormalities described above may be present in COVID-19 patients independently from the infection. Indeed, many patients developing a severe form of COVID-19 also exhibit important co-morbidities, such as chronic hypertension, diabetes, and chronic cardiovascular diseases. They may, therefore, display echocardiographic abnormalities before the infection and also be at particularly greater risk to develop further alterations in the setting of COVID-19, as abundantly reported in the literature [37,38,39]. It is also worth mentioning that performing an echocardiogram in COVID-19 patients exposes the echocardiographer to the risk of viral transmission [40]. It is, therefore, paramount to adhere to strict protocols of personal protective equipment and to limit the time of potential viral exposure [7]. A further important point to consider is that echocardiography may be technically challenging in these patients, notably due to respiratory distress, as well as mechanical ventilation. It is, indeed, well known that the quality of transthoracic echocardiographic images can be impaired by mechanical ventilation and suboptimal patient positioning in the ICU, as well as pulmonary disease itself [41]. Technical issues may be of particular concern for strain imaging, where the quality of images is essential for accurate strain tracking [42], which underlies the need of skilled and experienced echocardiographers to properly assess RV function in these complicated patients [43].

**Table 2 jcm-10-02535-t002:** Two-dimensional speckle tracking echocardiographic studies in COVID-19 patients.

N, Age, % Male	Patients	Echo Data	Pulmonary Circulation	Main Prognostic Findings	Ref
29 (64) 70%	ICU	▪ ACP: 41% ▪ ↓ RV LSF and RV FWLS in ACP pts	▪ Not reported	▪ Not reported	Beyls et al. [44]
30 (61) 65%	ICU	▪ ↓ TAPSE, S′: 5% ▪ ↓ RV GLS: 25% pts	▪ Not reported	▪ Poor correlation between usual RV indices and RV GLS	Gonzalez et al. [45]
35 (72) 79%	ICU	▪ RV GLS < −20% associated with mortality ▪ LV normal	▪ Not reported	▪ Mortality: 44% ▪ RV GLS not influenced by MV ▪ RV GLS potent independent predictor of mortality	Stockenhuber et al. [46]
12 (57) 42%	ICU	▪ RV and LV strain altered in all pts	▪ Not reported	▪ Adverse outcome (intubation, death): 40% ▪ RV abnormalities correlated with outcome	Krishnamoorthy et al. [47]
32 (56) 66%	ICU	▪ RV Dilat: 44% ▪ ↓ RV FWLS: 66%	▪ sPAP > 35 mm Hg: 42% ▪ PE: 16%	▪ Not reported ▪ Abnormal RV FWLS associated with LV systolic and diastolic Dys	Gibson et al. [48]
100 (59) 40%	ICU 22%	▪ ↓ RV LS and LV GLS in severe pts	▪ sPAP higher in severe pts	▪ Mortality: 22% (severe 50%, nonsevere 0%) ▪ RV GLS, LV GLS associated with mortality	Baycan et al. [49]
128 (61) 48%	ICU 15%	▪ ↑ TnI: 21.1% ▪ ↓ 3D RV EF, RV FWLS, RV FAC in non survivors ▪ LV not different	▪ PA larger in non survivors	▪ Mortality: 14%, correlates with RV Dys ▪ Echographic data associated with mortality: ▪ 3D RVEF < 42.5% ▪ RV FWLS > −18.9 ▪ RV FAC < 42.7% ▪ Larger RV and RA size	Zhang et al. [36]
120 (61) 57%	ICU 21%	▪ Non survivors: ▪ ↓ RV LS ▪ ↓ RV FAC ▪ ↓ TAPSE	▪ Non survivors: ▪ PA larger ▪ sPAP higher	▪ Mortality: 15% ▪ Best prognostic indicator: RV LS > −23% ▪ Mortality associated with RV Dys	Li et al. [34]
49 (66) 63%	ICU	▪ ↓↓ RV FWLS, RV LS in non survivors ▪ Cut-offs for death prediction ▪ −13.5% (RV LS) ▪ −18% (RV FWLS)	▪ Non survivors ▪ ↑ sPAP ▪ ↓ TAPSE/sPAP	▪ Mortality: 33%, independently correlated with RV Dys	Bursi et al. [50]
214 (69) 55%	Non ICU	▪ COVID vs. non COVID ▪ ↓ GLS, ↓ RV LS, ↓ TAPSE	▪ Not reported	▪ Mortality: 12% ▪ TAPSE, RVLS, GLS independent predictors	Lassen et al. [35]

Abbreviations: 3D: three-dimensional. ACP: acute cor pulmonale. EF: ejection fraction. FAC: fractional area change; FWLS; free wall longitudinal strain. GLS: global strain. ICU: intensive care unit. LS: longitudinal strain. LV: left ventricle. MV: mechanical ventilation. PA: pulmonary artery. Pts: patients. RV: fight ventricle. RV Dys: RV systolic dysfunction. sPAP; systolic pulmonary artery pressure. TAPSE: tricuspid annular plane systolic excursion.

To summarize, there is now substantial evidence that RV dysfunction is a hallmark pathophysiological alteration associated with COVID-19, as extensively reviewed by Dandel [51]. RV dilation occurs in up to 49% of patients, while RV systolic dysfunction occurs in up to 40%. As mentioned above, a particular form of *radial* RV dysfunction may even be observed in up to 70% of patients [21]. RV alterations are much more common than LV abnormalities and are associated with elevated levels of biomarkers of cardiac injury/dysfunction (TnI and BNP) and thrombotic/inflammatory biomarkers (D-dimer, C-reactive protein). Additionally, such alterations correlate with the importance of lung disease and the presence of higher PA pressure [52], which indicates that RV dysfunction in COVID-19 is an indicator of greater disease severity. Indeed, RV dysfunction in COVID-19 bears significant prognostic value, and it has been calculated that every 1 mm decrease of TAPSE is associated with a 20% increase mortality in COVID-19 patients [53].

## 4. Pathophysiology of RV Dysfunction in COVID-19

### 4.1. Physiology and Pathophysiology of the Right Ventricle

The inability of the RV to support optimal circulation without excessive use of the Frank–Starling mechanism (increase in stroke volume associated with increased preload) defines RV failure [4,54]. Therefore, RV dilation and systolic dysfunction are characteristic findings observed under conditions of *RV failure*. However, it must be underscored that a dilated RV may present a normal systolic function, thanks to various adaptive mechanisms, such as hypertrophy and homeometric autoregulation (see below). Such adaptations may be estimated using a recently described echocardiographic index, the RV load-adaptation index, using the tricuspid regurgitation velocity–time integral (VTI_TR_), and the measurement of the RV end-diastolic area (A_ED_) [55]. These notions imply, therefore, that RV dilation, RV dysfunction, and RV failure represent distinct, but not interchangeable, features, of RV pathophysiology [56].

The maintenance of normal RV function requires that it remains constantly adapted to the hydraulic load imposed by the pulmonary circulation, which defines the concept of RV-PA coupling, indicating the efficiency of energy transfer from the RV to the pulmonary circulation [56]. RV-PA coupling is best described by the ratio of RV maximal elastance (Ees) to arterial elastance of the PA (Ea) [57]. Ees represents a load independent measure of RV contractility, dependent on RV muscle mass and on the contractile properties of cardiomyocytes, whereas Ea is a representation of the hydraulic load of the pulmonary circulation [58]. This is schematically represented on the RV pressure–volume curve (Figure 1). Ees can be determined as the slope of the end-systolic PV relationship (Ees = RV end-systolic pressure/end-systolic volume), while Ea is depicted as the slope of the relationship between RV end-systolic pressure and RV stroke volume. Optimal coupling is indicated by a Ees/Ea ratio > 1, implying efficient RV to PA energy transfer. In the clinical setting, RV-PA coupling can be evaluated by the ratio of one surrogate determinant of RV systolic elastance, such as RV stroke work [59], TAPSE, or RV FAC [60], to one surrogate determinant of PA elastance, such as PA pressure [21,25].

The PA hydraulic load involves two main components, the first one being pulmonary vascular resistance (PVR), defined as (mean PAP-pulmonary artery wedge pressure)/cardiac output, the second one being pulmonary vascular capacitance (or PA compliance, PAC), defined as the ratio of the stroke volume (SV) to the PA pulse pressure (SV/PP). PVR represents the steady component, whereas PAC represents the pulsatile component of the PA hydraulic load, which must be met by adequate RV mechanical power, including both a steady and an oscillatory component [61]. An important feature of the pulmonary circulation is that PVR and PAC are proportionally and inversely correlated, so that their product (PVR × PAC, the pulmonary arterial time constant or RC-time) is constant. This implies that a marked decrease in PAC occurs when PVR is only slightly increased, and a reduction of PAC may, therefore, represent an important early marker of pulmonary vascular disease and increased RV afterload [57], notably in conditions of left heart disease [62].

When compared to the left ventricle, the RV is characterized by a greater diastolic compliance and a lower systolic elastance, implying that it is mostly adapted to changes in preload but not afterload. Under conditions of a progressive chronic increase in afterload (pulmonary hypertension), RV-PA coupling can be maintained by progressive RV hypertrophy. In contrast, a sudden increase in RV afterload cannot be tolerated (afterload mismatch) beyond the limited, transient, adaptation provided by the Anrep mechanism (reflex increase of contractility following an acute afterload increase, known as the cardiac homeometric autoregulation), and the RV must dilate to cope with the increased afterload (Frank–Starling mechanism, known as the cardiac heterometric autoregulation) [54,56]. As a result, the RV systolic function decreases, while its RV filling pressure increases, resulting in increased wall stress (hence, increased oxygen demand) and reduced cardiac output. The latter decreases further as a consequence of the leftward shift of the interventricular septum and decreased LV filling, leading to arterial hypotension, which further precipitates RV dysfunction by decreasing coronary perfusion pressure [56].

### 4.2. Mechanisms of RV-PA Uncoupling in COVID-19

A panoply of mechanisms has been proposed to account for the common observation of RV dysfunction in COVID-19. These can be subdivided into mechanisms increasing the PA hydraulic load and those reducing the contractile performance of the RV, both having the potential to promote RV-PA uncoupling. These mechanisms are summarized in Figure 2.

#### 4.2.1. Increase of the Pulmonary Hydraulic Load

##### Pulmonary Vascular Obstruction

Since the very early descriptions of COVID-19, a strikingly high incidence of thrombotic complications has been described, notably, affecting the pulmonary circulation. According to a recent systematic review of 28 studies totaling 2928 ICU COVID-19 patients, thrombotic complications occurred in 34% of patients, including 12.6% with pulmonary embolism (PE) [63]. In a recent Dutch survey of 947 patients, including 358 ICU patients, the cumulative incidence of venous thromboembolic complications after 30 days amounted to 19% in non-ICU and 23% in ICU patients [64]. Although the risk of PE is also increased in patients with pneumonia and ARDS unrelated to COVID, a significantly higher proportion of COVID-19 patients develop this complication, as notably reported by Helms et al. [65]. These authors compared two matched cohorts of COVID vs. non-COVID ARDS and found an incidence of PE of 11.7 vs. 2.1%, respectively (OR 6.2, *p* < 0.01) [65]. It is also particularly noticeable that pulmonary perfusion defects, assessed by dual energy CT scan (DECT), are observed in a majority of patients with COVID-19 ARDS [66].

In addition to this high incidence, COVID-19 associated PE frequently develops in spite of anticoagulant thrombo-prophylaxis [64,67], is characterized by a lower thrombus load and a more peripheral distribution than non-COVID PE [67,68], and combines both pulmonary arterial and venous thrombotic manifestations [69]. Furthermore, pathological examination of autopsy specimen of COVID-19 ARDS patients revealed the presence of microthrombotic obstruction of small pulmonary vessels, which were nine times more prevalent than in other forms of ARDS [70,71], and which displayed features of platelet-rich thrombotic microangiopathy [72]. Overall, these observations indicate that COVID-19 triggers a specific phenotype of vascular involvement, characterized by a distinct pattern of coagulopathy and immune-thrombotic alterations [73]. In this respect, some authors have proposed the concept of microCLOTS (microvascular COVID-19 lung vessels obstructive thrombo-inflammatory syndrome) [74] or the concept of diffuse pulmonary intravascular coagulopathy [75] to describe these unique abnormalities of the lung circulation.

The hypercoagulable state associated with COVID-19 has been termed COVID-19-associated hemostatic abnormalities (CAHA) or COVID-19 associated coagulopathy (CAC) [76,77]. It has many similarities with disseminated intravascular coagulation (DIC), and it is considered, indeed, a thrombotic phenotype of DIC, as it is usually not associated with bleeding, in contrast to conventional DIC [76]. It is, most notably, characterized by an elevation of plasma D-dimer, which signals ongoing fibrinolysis and correlates with the overall thrombus burden [77]. D-dimer elevation is an important prognostic indicator that may be used to grade the severity of COVID-19, according to a staging system proposed by the International Society on Hemostasis and Thrombosis (Stage I: <3× upper reference level of D-Dimer; Stage 2: 3–6×; Stage 3; >6×), together with other biomarkers (platelet count, fibrinogen, and activated partial thromboplastin time) [78]. Furthermore, as stated earlier, D-dimer elevation is associated with the development of RV dysfunction, pointing to the critical role of thrombotic occlusion of the pulmonary circulation in increasing RV afterload in COVID-19.

Several intricate mechanisms concur to foster the development of a prothrombotic state in COVID-19, which have been extensively reviewed in the recent literature [72,77]. These mechanisms include inflammation, endotheliopathy, thrombocytopathy, and complement activation [72]. Viral entry within the lung parenchyma triggers a local innate immune response with increased inflammatory cytokines and leukocyte recruitment [79], while at the same time, it directly activates the extrinsic coagulation system and suppresses plasminogen activation in lung epithelial cells [80]. Dysregulation of this initial response may result in a hyperinflammatory state, commonly known as a *cytokine storm*, promoting the development of ARDS and distant organ involvement. Cytokines, such as IL-1β, TNFα, IL-6, and chemokines, such as IL-8, activate coagulation by upregulating the expression of tissue factor and impairing fibrinolysis [79], by activating platelets, and by recruiting inflammatory cells. These events not only favor microthrombotic phenomena, but also promote the disruption of the normal anti-inflammatory and anticoagulant activity of the endothelium (endothelial dysfunction) [72,77,81,82].

The subsequent endothelial expression of adhesion molecules and inflammatory cytokine perpetuates this initial inflammatory response, notably, by upregulating the adhesion and activation of neutrophils, resulting in the release of damaging free radicals and neutrophil extracellular traps (NETs) [77]. NETs are major triggers of thrombotic complications, by promoting platelet activation and activating factor XI, and recent findings support a key role of NETs in the immune-thrombotic complications associated with COVID-19 [83,84]. Endothelial damage and intravascular coagulation are further exacerbated by an unrestrained activation of the complement cascade, in part secondary to SARS-CoV-2 interactions with the lectin pathway of complement activation [72], as well as by platelet hyperactivation linked to hypoxia, oxidative stress, platelet auto-antibodies, and direct platelet infection by SARS-CoV2 [72].

Endotheliopathy is at the core of COVID-19 pathophysiology, as evidenced by the increased circulating levels of various biomarkers of endothelial injury, including von Willebrand factor, angiopoietin2, thrombomodullin [72], endothelial-derived extracellular vesicles [85], and endothelial progenitor cells [86], whose levels directly correlate with the severity of COVID-19 [72,79]. In addition to the above immune-mediated mechanisms, direct SARS-CoV2 infection of the endothelium may significantly contribute to endotheliopathy. Endothelial cells express the ACE2 receptor required for viral entry, and several electron microscopy studies have, indeed, revealed the presence of SARS-CoV2 viral particles within endothelial cells from various organs, including the lung [82,87], which may result in endothelial cell death through apoptosis and pyroptosis [77]. Furthermore, ACE2 internalization and degradation following viral entry may result in altered angiotensin II (ATII) metabolism through ACE1/ACE2 imbalance, favoring ACE1-dependent AT II-dependent pro-inflammatory vasoconstrictor and procoagulant signaling at the expense of ACE2-dependent Angiotensin 1-7 anti-inflammatory vasodilator and anticoagulant signaling [81,82,88].

To summarize, macro- and microvascular thrombotic complications represent major pathophysiological features of COVID-19, resulting from a particular form of coagulopathy elicited by immune-mediated processes, endothelial damage and dysfunction, complement activation, and platelet hyperactivation. The lungs are at the epicenter of such abnormalities, with pulmonary vascular obstruction demonstrated in a significant proportion of patients, especially those suffering from ARDS, with a prevalence significantly higher than in other causes of ARDS. Such coagulopathy is typically associated with a marked elevation of plasma D-dimer, whose levels correlate with the severity of the diseases, as well as with the development of RV dysfunction consecutive to the increased pulmonary vascular hydraulic load.

##### Disturbances of Pulmonary Vasomotor Tone

Hypoxic vasoconstriction

The phenomenon of hypoxic vasoconstriction (HPV) is a physiological mechanism triggered by alveolar hypoxia, allowing redirection of lung perfusion from poorly to better ventilated alveoli, thereby reducing intrapulmonary shunt [89]. It relies on complex mechanisms based on cellular redox and bioenergetics changes, leading to the inhibition of K^+^ channels, Ca^2+^ influx, modulation of specific protein kinases, and subsequent vasoconstriction [90]. Although essential to limit the development of hypoxemia, widespread HPV in the setting of ARDS may promote pulmonary hypertension and increased RV afterload [91]. Such a mechanism is also noticeable in high altitude pulmonary edema (HAPE), where pulmonary hypertension coexists with pulmonary edema as a result of diffuse HPV [92].

While HPV is recognized as an important mechanism of increased pulmonary tone in non-COVID ARDS [91], its role in COVID-19 ARDS is currently controversial. Indeed, the latter may present as two distinct phenotypes, characterized by low or high elastance [93]. In the former, it has been claimed that inefficient HPV could favor massive intrapulmonary shunts in the affected lung areas, leading to profound hypoxemia with minimal hypoxemia-related symptoms, as reported in 32% of COVID-19 patients [94]. In such conditions, the infusion of the pulmonary vasoconstrictor almitrine has been reported to significantly improve oxygenation in 2/3 of patients [95]. Differences in the degree of HPV between COVID and non-COVID ARDS may also explain the findings by Caravita et al., who compared the hemodynamic profile of COVID vs. non-COVID ARDS [96]. Interestingly, pulmonary vascular resistance was lower in the group of COVID-19 patients, which was suggested to reflect a blunted HPV in this setting. Obviously, additional studies are needed to gain further insights into the precise role of HPV in the abnormalities of the pulmonary circulation in COVID-19.

2.Hypercapnic acidosis

Pulmonary vasoconstriction is a well-known consequence of hypercapnia [91], which seems to be primarily related to the effects of hypercapnia on extracellular H^+^ concentration, rather than to the elevated PCO_2_ itself [90]. In addition, hypercapnic acidosis may, under some circumstances, enhance HPV and contribute to increase pulmonary vascular tone in hypoxic conditions. Mechanisms are unclear and probably involve the release of arachidonic acid products, alteration of K^+^ channel activity in arterial smooth muscle, and activation of the sympathetic nervous system [90]. The detrimental effects of hypercapnia on RV afterload have been well demonstrated in the field of non-COVID ARDS, where a PaCO_2_ value > 48 mm Hg has been associated with the development of acute cor pulmonale and increased mortality [97]. This is of particular concern, owing to the frequent occurrence of hypercapnia (so-called “permissive hypercapnia”) in the setting of lung protective ventilation with low tidal volumes applied in ARDS [98], and efforts should, therefore, be put forward to limit the extent of hypercapnia in such conditions [91].

3.Angiotensin II-mediated vasoconstriction

As previously mentioned, SARS-CoV2 infects cells through the ACE2 receptor, which may result in an ACE1/ACE2 imbalance and lead to increased ATII signaling through the AT1 receptors, with parallel reduction of Angiotensin 1–7, which normally acts on the MAS receptor to oppose the actions of ATII [88]. Unopposed ATII signaling may promote vasoconstriction, vascular inflammation, microvascular thrombosis, and pulmonary vascular remodeling [99]. Interestingly, ACE2 downregulation has been associated with human pulmonary hypertension [100], and in the experimental setting, ACE2 protects from severe acute lung failure in various models of lung inflammation [101]. The role of ACE1/ACE2 imbalance and ATII signaling in COVID-19 remains, however, to be clarified. Indeed, while early data indicated a positive correlation between circulating ATII levels and the severity of the disease [102], two recent studies reported contrasting results with lower circulating ATII in non-survivors [103,104]. These observations suggest that severe COVID-19 is associated with global RAS dysregulation but not simply with a skewed balance in favor of ACE1 activity. This might reflect a global reduction of ACE1 and ACE2 expression in the damaged lung [103,104], a hypothesis which will deserve confirmation.

4.Vasoactive mediator imbalance

A critical physiological role of the pulmonary endothelium is to produce a balanced number of vasoactive mediators, permitting the maintenance of vasodilator tone and low vascular resistance in the pulmonary circulation. Two crucial mediators involved in this process are nitric oxide (NO, vasodilator) and endotelin-1 (vasoconstrictor) [105]. The severe endothelial dysfunction associated with the endotheliopathy of COVID-19 likely affects this balance, as previously shown in non-COVID ARDS, where reduced formation of NO concurrent to increased formation of endothelin-1 have been reported [106]. Recent studies have indicated that NO availability is markedly reduced in the lungs of COVID-19 patients, as a result of decreased endothelial synthesis and increased degradation by reactive oxygen species [107]. Accordingly, inhaled NO therapy has been proposed to reduce pulmonary vascular resistance in COVID-19 ARDS, and several clinical trials on this topic are currently ongoing [108].

5.Additional mechanisms of increased pulmonary vascular tone

In addition to their effects in promoting thrombotic complications in the lung microcirculation, cytokines, such as IL-6 and TNFα, may also trigger vasoconstriction, both directly and indirectly by suppressing bone morphogenetic receptor 2 signaling in pulmonary vascular smooth muscle cells [99,108]. These cytokines also upregulate the formation of hyaluronan, a component of the extracellular matrix that may represent an important mediator of lung damage in COVID-19 [109] and can promote pulmonary hypertension by fostering stiffness and proliferation of vascular smooth muscle cell [110].

##### Mechanical Ventilation

Positive pressure mechanical ventilation (MV) introduces significant modifications in the physiology of heart–lung interactions, which include both constant effects related to the application of positive end expiratory pressure (PEEP) and cyclic changes related to positive pressure inspiration. The subsequent increase in *pleural pressure* (intrathoracic pressure) exerts opposite actions on the loads of both ventricles, reducing RV preload while increasing LV preload [111]. Additionally, positive pressure inspiration elevates alveolar pressure (the *plateau pressure*), thereby increasing the *transpulmonary pressure* (alveolar minus pleural pressure) and RV afterload, while it increases further pleural pressure, reducing LV transmural pressure and LV afterload [91,112]. The transpulmonary pressure (*lung stress*) and the alveolar deformation at each tidal ventilation (*lung strain*, partly evaluated by the *driving pressure*, which is the plateau pressure minus PEEP) are important components of the amount of energy transmitted to the lung parenchyma during MV. Too high lung stress and strain, hence, too high energy transfer to the lung, are key determinants of so-called ventilator-induced lung injury [113]. Therefore, current recommendations indicate that the transpulmonary pressure (approximated by the plateau pressure) should remain below 30 cm H_2_O and that tidal volume should be maintained at around 6 mL/kg in order to keep the driving pressure below 15 cm H_2_O during MV of ARDS [114].

Since the transpulmonary pressure reflects the extramural pressure of alveolar vessels, lung overdistension during MV will have a major impact on RV afterload by compressing these vessels, increasing pulmonary vascular resistance (PVR). In contrast, at very low lung volumes, the increased elastic recoil forces of the lung promote the collapse of extra-alveolar vessels, which also results in an increase in PVR. This explains the U-shaped relationship between PVR and lung volumes and indicates that PVR is minimal at the functional residual capacity [98]. In ARDS, the reduced lung compliance, the requirements for relatively high PEEP levels, and the inhomogeneity of lung involvement, may expose patients to a significant risk of overdistension and abnormally high transpulmonary and driving pressure. In turn, these effects will increase significantly the pulmonary vascular hydraulic load and jeopardize the already overloaded RV [91]. It has been shown that maintaining a plateau pressure below 27 cm H_2_O [115] and a driving pressure below 18 cm H_2_O [97] is crucial to lower the risk of RV dysfunction and acute cor pulmonale in non-COVID ARDS, and the same precautions should be applied in COVID-19 ARDS.

#### 4.2.2. Reduction of Right Ventricular Contractility

##### Evidence of Myocardial Injury in COVID-19

The notion of myocardial involvement in COVID-19 emerged already in the very early descriptions of the disease. In several cohort studies from China, myocardial injury, as defined by elevated values of serum TnI, was reported in 7–28% of patients and was significantly associated with severe forms of the disease, the levels of inflammatory biomarkers, and mortality [38,116,117]. An elevation of cardiac biomarkers, including TnI and natriuretic peptides, has been reported in numerous cohort studies, with a prevalence reaching up to 50%, most notably in ICU-hospitalized patients [118,119,120] and with considerable prognostic value (OR for mortality 6.64, *p* = 0.03), according to a recent meta-analysis [121].

A variety of clinical manifestations associated with cardiac injury has been reported, including right and left ventricle dysfunction, acute heart failure and cardiogenic shock, myocardial ischemia, ventricular arrhythmias, pericardial effusions, and stress cardiomyopathy [122], which have been grouped under the acronym ACovCS (acute COVID-19 cardiovascular syndrome) [120]. Furthermore, a series of cardiac magnetic resonance imaging studies have indicated some forms of myocardial involvement in a significant proportion of COVID-19 patients, including asymptomatic and recovering patients. The most frequent reported abnormalities include myocarditis (40%), myocardial edema (51%), and late gadolinium enhancement (43%) pointing to some forms of myocyte necrosis [123]. In spite of their prevalence, it is noticeable that the majority of findings reported were mild, and their clinical significance remains so far unknown [122].

##### Potential Mechanisms of Cardiac Injury and Dysfunction in COVID-19

Myocardial inflammation

Various cardiac cells express the ACE2 receptor and could be, therefore, targets for SARS-CoV2 infection, including cardiomyocytes, cardiac fibroblasts, pericytes, and possibly endothelial cells (but the latter is controversial) [124,125]. Recent studies using electron microscopy, RNA sequencing, and direct infection of pluripotent stem-cell-derived cardiomyocytes, as well as human engineered heart tissue, have definitely proven the capacity of SARS-CoV2 to infect, replicate, and induce cell death in cardiomyocytes [126]. In spite of this evidence, the actual incidence of typical myocarditis (as diagnosed by the presence of at least seven CD3-positive T-lymphocytes/mm^2^) in COVID-19 is extremely uncommon, occurring in only 4.5% of highly selected cases undergoing autopsy or endomyocardial biopsy (EMB) [124]. In contrast to the paucity of data regarding lymphocytic myocarditis in COVID-19, there is a significant proportion of autopsy [127] and EMB [128] specimens displaying an increased interstitial macrophage infiltration of the myocardium. Such high levels of myocardial macrophages point to non-specific myocardial inflammation resulting from high systemic levels of cytokines instead of direct viral-mediated myocarditis [127,129], as reported in other forms of pneumonias, although the occurrence of myocardial injury appears significantly higher in COVID than non-COVID pneumonia [120].

2.Myocardial ischemic injury

While coronary plaque rupture leading to ST-segment elevation myocardial infarction has been occasionally reported in COVID-19 [130], emerging evidence indicates that microcirculatory derangements may represent a more frequent mechanism of myocardial ischemia. This has been well documented in several histopathological studies demonstrating a high prevalence of microvascular thrombosis, associated with focal areas of necrosis [131,132]. The mechanisms responsible for these alterations are presumably similar to those previously discussed, including coagulopathy, endotheliopathy, and platelet and complement activation [122]. Indeed, a recent study identified a unique composition of COVID-19 associated coronary microthrombi that were significantly enriched in fibrin and terminal complement [131]. These findings provide a pathological basis for the reported occurrence of myocardial infarction with non-obstructive coronary arteries (MINOCA) in COVID-19 patients [130,133]. In addition to obstructive and non-obstructive coronary hypoperfusion, further mechanisms may precipitate myocardial ischemic injury, including low arterial blood pressure and hypoxemia, as well as increased myocardial demand due to tachycardia, fever, and increased RV wall stress due to volume and pressure overload.

3.Dysregulated RAS and inflammatory cytokines

As discussed previously, the increased degradation of ACE2 in response to SARS-CoV2 infection may result in a theoretical unopposed ATII signaling [122] with negative actions on the heart, comprising inflammation, oxidative stress, negative inotropic effects, and adverse remodeling [134]. Whether ATII contributes to myocardial damage and dysfunction in COVID-19, and whether RAS-interfering drugs have therapeutic advantages, remain speculative so far and are the focus of several ongoing clinical investigations. In addition to ATII, several inflammatory cytokines known for their direct negative inotropic actions, most notably, IL-1β, IL-6, and TNFα, may also contribute to impair cardiac contractility in severe COVID-19. In this respect, it will be essential to evaluate the impact of anti-cytokine therapies on indices of cardiac dysfunction in COVID-19 (e.g., tocilizumab, canakinumab).

## 5. Treatment of COVID-19-Associated Right Ventricular Dysfunction

Treatment of RV dysfunction in COVID-19 must follow the usual recommendations for acute RV failure of any etiology, including maintenance of perfusion pressure, preload optimization, reduction of afterload, and inotropic support. In addition, some specific aspects related to the role of systemic inflammation and coagulopathy in COVID-19 must be considered.

### 5.1. Anticoagulation

As discussed previously, COVID-19 exposes patients to significant risks of macro- and microthrombotic complications, most significantly in patients with the most severe forms of the disease. While therapeutic anticoagulation is mandatory in patients with documented thromboembolic events, strategies for antithrombotic prophylaxis remain not completely defined. According to ISHT interim guidance, standard prophylaxis with low molecular weight heparin (LWMH) or unfractioned heparin in patients with renal failure (creatinine clearance < 30 mL/min) is recommended for all hospitalized patients with COVID-19, except in patients with active bleeding or platelet counts below 25 × 10^9^/L [135]. In patients requiring high levels of oxygen support, exhibiting severe inflammation, and displaying significant increases in D-dimers, intermediate or therapeutic doses of LMWH or unfractioned heparin are proposed by several authorities [136,137]. A great number of ongoing clinical trials are presently evaluating several antithrombotic regimens, including anticoagulants and antiplatelet and fibrinolytic agents in COVID-19, as presented in a recent extensive review [138].

### 5.2. Anti-Inflammatory Therapies

Systemic inflammation is at the core of the pathophysiology of severe COVID-19 and is instrumental in the development of RV dysfunction. The RECOVERY trial [139] reported a significant reduction of 28-day-mortality with the use of up to 10 days dexamethasone (6 mg/day) in patients mechanically ventilated or requiring supplemental oxygen, and this therapy is now recommended for this category of patients. Moreover, both the REMAP-CAP trial [140] and the RECOVERY trial (preprint: doi.org/10.1101/2021.02.11.21249258, accessed on 3 June 2021) provided additional evidence that the anti-IL-6 receptor monoclonal antibody tocilizumab (at a single iv dose of 8 mg/kg) offered additional survival benefits in severely ill patients, requiring high levels of supplemental oxygen or mechanical ventilation and displaying increased circulating biomarkers of inflammation (CRP > 75 mg/L). According to the most recent NIH recommendations (https://www.covid19treatmentguidelines.nih.gov, accessed on 3 June 2021), tocilizumab should, therefore, be used in combination with dexamethasone in this subset of severely ill COVID-19 patients. How these anti-inflammatory regimens influence the heart, and especially RV function, remains, however, to be documented.

### 5.3. Specific Management of RV Failure

Perfusion pressure

The maintenance of adequate systemic arterial pressure is crucial to ensure coronary perfusion of the overloaded RV (Figure 3). Under normal conditions, the RV is perfused both in systole and diastole, but increased RV afterload and pulmonary systolic pressure jeopardizes systolic perfusion, whereas increased end-diastolic pressure reduces the diastolic perfusion gradient [4,141]. Vasopressor therapy is, therefore, mandatory to maintain systemic arterial pressure above pulmonary arterial pressure. Norepinephrine is used as a first line agent but should be administered with caution owing to its ability to raise pulmonary vascular resistance at doses > 0.5 μg/kg/min and induce tachycardia [141,142]. Vasopressin is a useful alternative (at doses < 0.03 U/min) owing to its much greater action in raising systemic rather than pulmonary vascular resistance and its lack of chronotropic actions [4,142]. The main effects and adverse effects of vasopressors and inotropes (discussed below) used to increase RV perfusion pressure are summarized in Table 3.

2.Volume management

Volume overload of the RV reduces LV filling by septal interactions and pericardial constraint, increases RV wall stress and oxygen consumption, and favors tricuspid regurgitation by annular dilation [143]. All these effects must be avoided by proper fluid management. Stimulated diuresis (loop diuretics) or renal replacement therapies, in case of diuretic resistance or renal failure, should aim at maintaining the volume status in order to keep central venous pressure between 8–12 mm Hg [4].

3.Afterload reduction

The most important mechanism of acute RV failure in ARDS, and more specifically in COVID-19, is related to its inability to cope with an acutely increased pulmonary vascular hydraulic load, resulting in RV-PA uncoupling and RV dilation leading to acute cor pulmonale (ACP) [144]. Reversible causes include hypoxia, hypercapnic acidosis, and positive pressure mechanical ventilation. In a prospective study including 752 patients with moderate-to-severe Non-COVID ARDS, Mekontso-Dessap reported a prevalence of ACP of 22% [97]. Four independent variables were significantly associated with the risk of ACP, including: (a) pneumonia as a cause of ARDS; (b) driving pressure ≥ 18 cm H_2_O; (c) a ratio of PaO_2_ to inspired fraction of O_2_ (P/F O_2_) < 150 mmHg; (d) PaCO2 ≥ 48 mmHg. In addition, Jardin et al. found that the risk of ACP was lower when plateau pressure was maintained below 27 cm H_2_O [115].

These findings clearly indicate the need for a rigorous application of a protective ventilation strategy in ARDS, not only to reduce the risk of ventilator-induced lung injury but also of RV dysfunction [145]. When such conditions are insufficient to maintain adequate gas exchange, prone position ventilation should be applied, as it has been shown to improve oxygenation and reduce mortality in patients with the most severe forms of ARDS [146]. Importantly, prone position is associated with significant beneficial hemodynamic effects, notably, with respect to the RV. Indeed, by promoting lung recruitment, prone position reduces hypoxemia, hypercapnia, driving pressure, and plateau pressure, thereby reducing the pulmonary vascular hydraulic load and improving RV function [145]. Most recent guidelines have endorsed prone positioning in the clinical management of severely hypoxemic COVID-19 patients, both under spontaneous awake ventilation (self-proning) and mechanical ventilation [147].

In addition to protective ventilation and prone position, inhaled nitric oxide (iNO) is currently evaluated as a possible adjunctive therapy to improve oxygenation and reduce pulmonary vascular resistance in severe COVID-19. This therapy might also exploit the anticoagulant and antiviral properties of NO [107]. Such an approach makes sense owing to the severe endothelial dysfunction, which is expected to reduce the endogenous NO production and shift the vasoactive balance towards a more vasoconstrictor phenotype [107]. Expected effects of iNO include the reduction of PVR and an improvement of oxygenation via a reduction of shunt [148]. A few observational studies found controversial effects of iNO on oxygenation (improved or unchanged) [149,150], and there is only limited information suggesting improved pulmonary hemodynamics [149], especially in the most severe patients [150]. Table 4 summarizes the studies on iNO in COVID-19 published so far. The results of several ongoing prospective clinical trials are expected soon, which will permit us to better determine the place of iNO in the COVID-19 therapeutic arsenal [107].

4.Inotropic support

Inotropes are indicated when RV/PA uncoupling results in a fall of cardiac output. In addition to increasing RV (and LV) contractility, inotropes reduce RV volume and pressure overload. Ideally, the effects of therapy should be evaluated with invasive hemodynamic monitoring using a pulmonary artery catheter [4]. The first line inotrope is generally the β-adrenergic agonist dobutamine, but its use is limited by the risk of tachyarrhythmias and increased myocardial oxygen demand [4]. Milrinone, a phosphodiesterase inhibitor with both inotropic and vasodilator actions may be used instead of dobutamine, with the advantage to reduce pulmonary vascular resistance in addition to increasing RV contractility with less chronotropic actions [157]. However, milrinone should be avoided in the presence of hypotension owing to its vasodilator properties [4]. Finally, levosimendan, a calcium sensitizer and activator of K^+^ATP channels, may represent an interesting alternative to dobutamine. Although it may also induce hypotension, levosimendan has been shown to improve RV/PA coupling [158] and does not have the disadvantage of increasing myocardial oxygen demand [159]. In a pilot study of 35 non-COVID ARDS patients, levosimendan has been associated with a significant improvement of cardiac output, a decrease of pulmonary vascular resistance, and a reduction of RV volume overload [160]. A prospective randomized clinical trial recruiting 120 ARDS patients is currently under way to evaluate the benefits of levosimendan on pulmonary hemodynamics in this setting (NCT04020003). The main effects and side effects of inotropes used to increase RV contractility are presented in Table 3.

5.Extracorporeal membrane oxygenation (ECMO)

Veno–venous ECMO (VV-ECMO) may be used as a rescue therapy in refractory ARDS to improve gas exchange and favor ultra-protective lung ventilation [161]. In non-COVID ARDS, the largest randomized controlled trial (EOLIA trial) reported a 60-day mortality of 35% in patients treated with VV-ECMO, as opposed to 45% in patients under conventional treatment. Although the difference was not significant (*p* = 0.09), subsequent analyses of the trial, as well as a meta-analysis on ECMO in ARDS, lent support to a positive effect of VV-ECMO on the survival of ARDS patients [161]. These positive results led the World Health Organization (WHO), as well as the Extracorporeal Life Support Organization (ELSO) [162], to suggest that VV-ECMO might be considered in the treatment of the most severely ill COVID-19 ARDS patients [163]. Early reports on ECMO in COVID-19 reported unfavorable outcomes, with mortality rates up to 90% [164], raising concerns on the justification of this therapy in conditions of a global pandemic with major strains on health care resources [165]. However, improved patient selection and ECMO program organization over time allowed for obtaining much better results than initially reported, and the role of VV-ECMO in COVID-19 is constantly evolving with the continuous release of new data [166].

Although a detailed discussion on VV-ECMO in COVID-19 ARDS is beyond the scope of this review and can be found elsewhere [167], some results of the most recent large cohorts of patients will be briefly presented. In a retrospective analysis of 83 patients treated with VV-ECMO for severe COVID-19 ARDS (median PO_2_/FiO_2_ of 60 mm Hg before ECMO initiation), Schmidt et al. reported an estimated 31% probability of day-60 mortality, comparable to that of the EOLIA trial for non-COVID ARDS (35%) [168]. Lorusso et al. presented data gathered from 177 centers in Europe and Israel, representing a total of 1531 COVID-19 patients treated with ECMO, including 91% on VV-ECMO. The mean duration of ECMO support was 18 days and overall mortality 45% [169]. Barbaro et al. reported findings on 1093 patients from the ELSO registry, obtained from 213 hospitals in 36 countries. The median PO_2_/FiO_2_ in the 6 h preceding ECMO was 72 mm Hg, the median duration of ECMO support was 13.9 days, and 90-day mortality was 37.4% [163].

Further important results were recently published by Lebreton et al., who analyzed data from 302 COVID-19 patients treated with VV-ECMO in 17 ICUs in Paris [170]. The rate of 90-day survival was 46% in this cohort, and a multivariable analysis indicated that a longer time between intubation and ECMO, older age, and pre-ECMO renal dysfunction were independently associated with reduced 90-day survival. Furthermore, this study showed that survival was strongly associated with the center’s experience in managing VV-ECMO patients, which underscores the importance of performing VV-ECMO in high-volume expert centers and the need to apply centralization and regulation of ECMO indications [170]. In a very recent study on the outcome of VV-ECMO in COVID-19, the authors sought to estimate the effect of ECMO on mortality in comparison to conventional therapy. For this purpose, they analyzed data from 190 patients treated with ECMO among a cohort of 5122 critically ill COVID-19 patients in 68 US hospitals [171]. They performed a “target trial emulation”, in which only mechanically ventilated patients with an age < 70 y, without malignancy, hospitalized in an ECMO center, and a PaO_2_/FiO_2_ ratio < 100 mm Hg between day 1 and day 7, were included. Using this strategy, a significant survival benefit in favor of ECMO was noted over a median follow up of 38 days (34.6% vs. 46.1% mortality, HR 0.55, *p* < 0.001), suggesting that ECMO may reduce mortality in selected patients with severe respiratory failure from COVID-19 [171].

The growing body of literature on VV-ECMO in COVID-19 ARDS, therefore, suggests that this therapeutic strategy results in survival rates quite similar to that noted for VV-ECMO in non-COVID-19 ARDS. However, it must be stressed that VV-ECMO consumes significant resources, both in terms of equipment and intensive care health care providers, which must be critically taken into account in the context of a global pandemic with a very large number of ARDS patients and a scarce number of available resources. In addition, VV-ECMO is associated with an important rate of complications, most especially thrombotic and hemorrhagic, which appear to be more frequent in the setting of COVID-19 than in other forms of ARDS, notably, with a greater incidence of intracranial bleeding [170]. These considerations imply that very strict and unified criteria for VV-ECMO implementation in COVID-19 be applied together with stringent contraindications, and that well-coordinated ECMO referral centers be organized within geographic regions. All these aspects have been recently extensively reviewed by the ELSO, which has provided complete updated guidelines for the use of ECMO in COVID-19 patients with severe cardiopulmonary failure, to which we refer the interested reader [166].

With specific respect to the possible effects of VV-ECMO on the right ventricle, some points deserve further discussion. The rapid reduction of PaCO_2_ and improved oxygenation provided by VV-ECMO have been shown to promote an immediate decrease of PA pressure and RV unloading [172], which suggests that VV-ECMO could be particularly useful in COVID-19 patients with severe RV dysfunction [173]. In this regard, Mustafa et al. reported that VV-ECMO, using a single stage dual-lumen right atrium-to-pulmonary artery cannula, provided significant RV support in a cohort of 40 COVID-19 ARDS patients. Only 15% mortality was noted, and 73% of patients were discharged alive from the hospital while no longer receiving oxygen [174]. Joyce used a comparable approach in nine patients, treated with a percutaneous RV assist device coupled to an oxygenator (oxy-RVAD strategy), with only one reported death [175]. Obviously, the data by Mustafa and Joyce should be viewed as preliminary and interpreted with great caution, but they may suggest the interesting concept that supporting the RV in addition to provide extracorporeal gas exchange may be of particular benefit in the most severe COVID-19 patients. Clearly, this hypothesis will deserve future evaluation in larger scale clinical trials [173].

## 6. Conclusions and Future Perspectives

Although RV dysfunction is a known complication associated with various forms of ARDS, its prevalence appears particularly elevated in the setting of COVID-19 ARDS. Echocardiographic studies have shown that RV dysfunction in COVID-19 may take the form of a specific *radial*, instead of *longitudinal* dysfunction, and that it is commonly accompanied by RV dilation due to pressure overload. Complex mechanisms pertaining to dysregulated pulmonary circulation and impaired RV contractility explain the frequent occurrence of RV dysfunction in COVID-19. Furthermore, RV alterations generally reflect the severity of the disease and are associated with a dismal prognosis. A deep knowledge of the pathophysiological disturbances affecting the pulmonary circulation and the heart in COVID-19 is mandatory to propose the most appropriate therapies to restore the coupling between the RV and the pulmonary artery. In this respect, numerous ongoing and future clinical trials evaluating different anticoagulant regimens, novel anti-inflammatory strategies, inhaled pulmonary vasodilators, RAS-targeted drugs, various inotropic drugs, and innovative modes of extracorporeal support, will provide invaluable clues to improve the management of RV dysfunction complicating severe COVID-19.

## Figures and Tables

**Figure 1 jcm-10-02535-f001:**
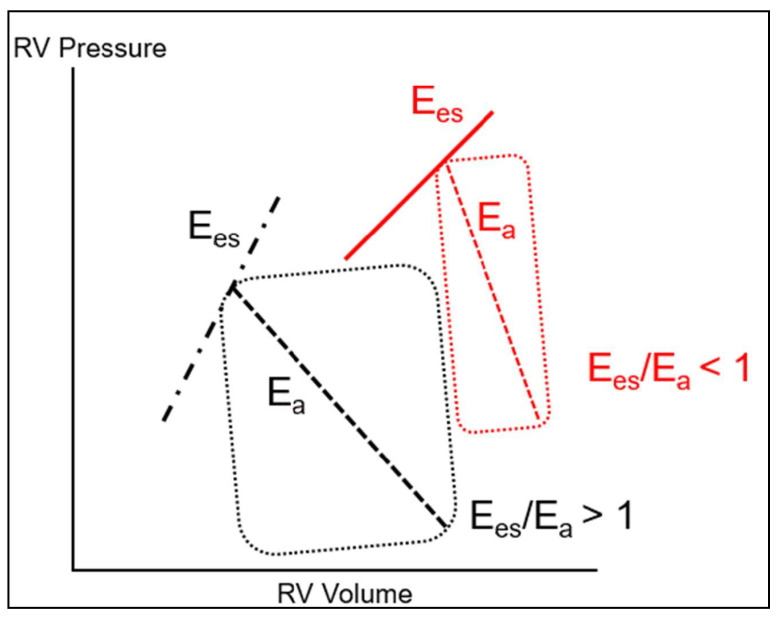
RV-PA coupling. The pressure volume curve of the RV allows characterization of the end-systolic PV relationship, whose slope corresponds to the RV systolic elastance (Ees), a load-independent measure of RV contractility. The line joining the RV end-systolic volume with its end-diastolic volume is termed the pulmonary arterial elastance (Ea), which is a measure of the afterload as it is seen by the RV. The ratio between Ees and Ea defines the RV-PA coupling, which should always be kept >1 for optimal RV efficiency (black lines). Increased Ea or/and reduced Ees precipitates RV uncoupling (Ees/Ea < 1). The RV must dilate (Frank–Starling mechanism) to maintain its output, at the expense of a marked increase in wall stress, hence myocardial oxygen demand (Red lines).

**Figure 2 jcm-10-02535-f002:**
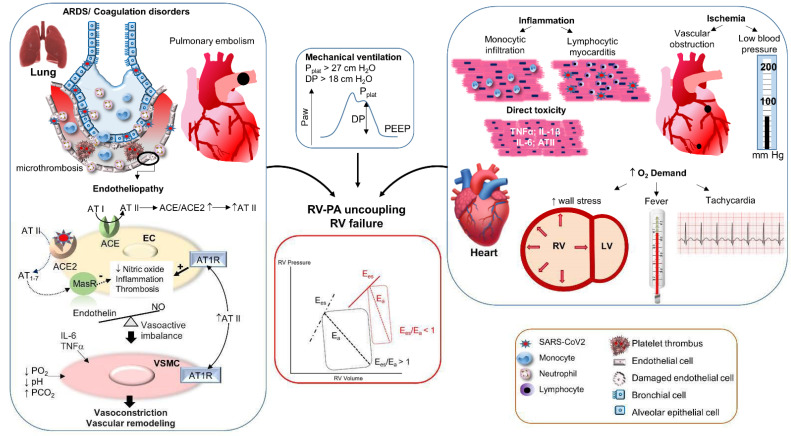
Pathophysiological mechanisms of RV dysfunction in COVID-19 ARDS. Upper left panel: Viral infection of lung epithelial cells triggers an intense inflammatory response, promoting the recruitment of inflammatory monocytes and neutrophils, as well as the upregulation of an array of inflammatory cytokines, resulting in endothelial damage and dysfunction, commonly defined as endotheliopathy. Lower left panel: The endotheliopathy is further aggravated by ACE/ACE2 imbalance following ACE2 degradation in endothelial cells (EC) in response to SARS-CoV2 receptor ligation. In turn, ACE2-dependent metabolism of angiotensin II (AT II) into Angiotensin_1-7_ (AT_1-7_) is reduced, with decreased signaling through the Mas Receptor (MasR), which normally conveys anti-inflammatory, vasodilatory (NO-dependent), and antithrombotic effects, and increased AT II signaling through the AT II receptor type 1 (AT1R). The latter promotes prothrombotic and pro-inflammatory effects, favoring diffuse pulmonary intravascular coagulation, with both micro and macrothromboses (pulmonary embolism), as well as vasoactive imbalance leading to sustained vasoconstriction. In addition, direct actions on vascular smooth muscle cells (VSMC) of AT II, cytokines (IL-6, TNFα), hypoxia, and hypercapnic acidosis amplify vasoconstriction. Middle upper panel: In patients requiring mechanical ventilation, too high plateau pressure (P_plat_ > 27 cm H_2_O) and driving pressure (DP > 18 cm H_2_O) may compress alveolar vessels. The overall resulting effects of these processes is an increase in pulmonary vascular resistance, which increases the pulmonary vascular hydraulic load. Right panel: COVID-19 may be complicated by myocardial inflammation due to direct viral infection (lymphocytic myocarditis, rare) or, more often, to a monocytic type of inflammation in the setting of systemic inflammation. Myocardial damage and reduced contractility may be further enhanced by the direct actions on cardiomyocytes of cytokines (TNFα, IL-1β, IL-6) and AT II, as well as by ischemic complications resulting from macro- or microvascular thromboses and systemic hypotension, which may be further aggravated in conditions of increased myocardial oxygen demand induced by fever, tachycardia, and increased wall stress (RV dilation). Inflammation and necrosis of cardiomyocytes, in turn, result in a decrease of RV contractility. Middle lower panel: The concomitant increase in pulmonary vascular hydraulic load (increased arterial elastance, Ea) and reduction in RV contractility (decreased RV end-systolic elastance, Ees) precipitate RV-PA uncoupling (Ees/Ea < 1), leading to RV dilation and acute RV failure (red-colored pressure-volume RV curve).

**Figure 3 jcm-10-02535-f003:**
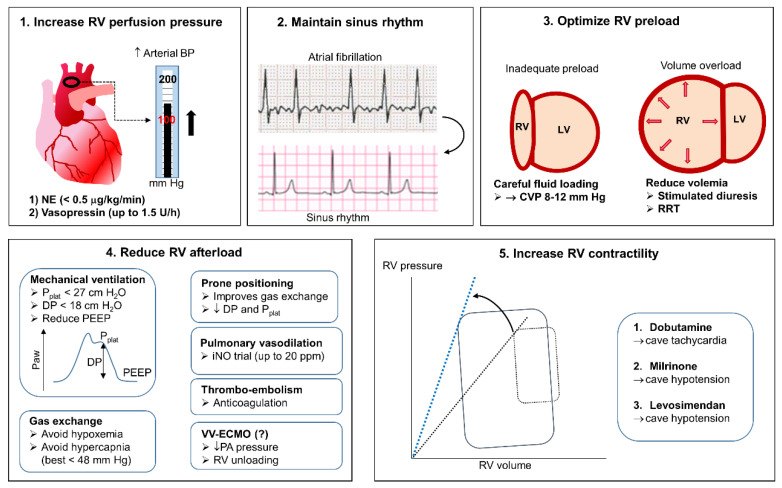
Specific management of RV failure. The figure illustrates the five critical components of the treatment of RV failure, including: (**1**) The increase of arterial blood pressure (BP) using vasopressors, including norepinephrine (NE, first line) and vasopressin (second line, when NE doses are >0.5 μg/kg/min). (**2**) The maintenance of sinus rhythm, using anti-arrhythmic drugs or direct current cardioversion in case of atrial fibrillation. (**3**) The optimization of RV preload, monitored by echocardiography and central venous pressure. The aim is to ensure adequate RV filling (target a CVP of 8–12 mm Hg) and to aggressively treat volume overload, as evidenced by dilated RV and paradoxical septum movement towards the left ventricle (LV) cavity, using diuretics or renal replacement therapy (RRT). (**4**) The reduction of RV afterload (a) by avoiding elevated plateau pressure (P_Plat_), driving pressure (DP), and positive end-expiratory pressure (PEEP) in mechanically ventilated patients; (b) by avoiding hypoxemia and hypercapnic acidosis; (c) by prone positioning the patient; (d) by administering inhaled nitric oxide (iNO); (e) by treating thrombo-embolic complications; and (f) in refractory cases, by discussing the possibility of extracorporeal support using veno–venous extracorporeal membrane oxygenation (VV-ECMO). (**5**) The increase of RV contractility using inotropic agents, including Dobutamine (first line) and /or Milrinone or Levosimendan (second line). See text for further details.

**Table 1 jcm-10-02535-t001:** Conventional echocardiographic studies in COVID-19 patients.

N (Age) % Male	Patients	Echography/Biomarkers	Pulmonary Circulation	Main Prognostic Findings	Ref
332 (66.9) 71.4%	ICU 22%	▪ ↑ TnI: 37% ▪ RV Dilat in pts with ↑ TnI	▪ PA > 32 mm associated with mortality	▪ 40.6% vs. 8.6% mortality ▪ in pts with ↑ TnI	Ferrante et al. [3]
4 (50–67 y) 75%	ICU	▪ RV Dilat + Dys ▪ No LV Dys	▪ Suspect PH in all pts	▪ Thrombolysis in 2 pts ▪ 75% mortality	Garcia-Cruz et al. [5]
5 (42–76 y) 60%	ICU	▪ Circulatory shock ▪ Severe RV Dilat + Dys	▪ Suspect PH in all pts	▪ 2 pts treated with thrombolysis survived	Creel-Bulos et al. [1]
66 (60) 57.6%	ICU 58%	▪ RV dilation: 74% (severe: 15%) ▪ RV Dys: 28% ▪ LVEF 59 ± 10% ▪ ↑ D-Dimers: 71% ▪ ↑ TnI: 62%	▪ ↑ PA pressure suspected	▪ Mortality: 38% ▪ No correlation between D-dimer—TnI and RV Dilat	Schott et al. [6]
29 (NA)	ICU	▪ Pericardial effusion: 17% (5 pts) ▪ RV Dilat: 10%	▪ Normal PA pressure	▪ Death in 2/5 pts ▪ Normalization of echo in 2/5 pts	Rauch et al. [2]
110 (66) 64%	ICU	▪ RV Dilat: 31%, 2/3 with ▪ RV Dys ▪ No LV alterations	▪ PE in 16% pts with RV Dilat	▪ Mortality 41% vs. 11% in pts with vs w/o RV Dilat ▪ RV Dilat only predictor of mortality (multivariate)	Argulian et al. [7]
49 (64.3) 54.3%	ICU	▪ RV Dys: 36% (ARDS)	▪ ↑ sPAP in ARDS	▪ No data	Li et al. [8]
100 (66.1) 63%	ICU	▪ RV Dilat/Dys: 39% ▪ LV systolic Dys: 10% ▪ LV diastolic Dys: 16% ▪ ↑ TnI: 20%	▪ ↑ PVR associated with RV Dys	▪ RV Dys/Dilat and low LVEF associated with mortality	Szekely et al. [9]
416 (47) 48%	ICU 8% Echo in 57 pts	▪ ↑ TnI: 51% ▪ Thickened LV: 39% ▪ ↓ LVEF: 16% ▪ RV Dilat: 10%	▪ PH: 29% (ICU pts)	▪ Mortality 10% (ICU)	Zeng et al. [10]
51 (63) 80%	Non-ICU	▪ ↑ TnI/BNP: 47% ▪ LV systolic Dys: 27% ▪ RV Dys: 10%	▪ PE: 27%	▪ 1 death ▪ No correlation echo/ biomarkers ▪ PE not associated with RV Dys	Van den Heuvel et al. [11]
45 (61) 51%	NA	▪ ↑ TnI: 18% ▪ ↑ BNP: 36% ▪ ↓ LVEF: 31% ▪ RV Dilat: 13%	▪ PH: 24% ▪ PE: 4.5%	▪ Not reported	Vasudev et al. [12]
115 (64) 60%	ICU	▪ ↑ TnI: 23% ▪ RV Dilat and Dys in pts with ↑ TnI ▪ LV normal	▪ sPAP > in pts with RV Dys	▪ Mortality: 50% vs. 8% with ↑ TnI ▪ RV Dys independently associated with mortality	D’Andrea et al. [13]
200 (62) 66%	non ICU	▪ RV Dys: 14.5% ▪ No LV Dys ▪ TnI/BNP ↑ with RV Dys and PH	▪ sPAP > 35 mm ▪ Hg in 12%	▪ PH associated with mortality (42% vs. 9%) ▪ RV Dys not associated with mortality	Pagnesi et al. [14]
98 (68) 77%	ICU 57%	▪ ↓ LVEF: 13% ▪ RV Dys: 14% ▪ RV Dilat: 46% ▪ TnI/NT-BNP correlated with LV and RV Dys	▪ NA	▪ 30 day mortality: 13% ▪ TnI, NT-proBNP, LV and RV Dys associated with mortality	Rath et al. [15]
72 (18–80 y) 72%	ICU 56%	▪ RV Dys: 40% ▪ RV Dilat: 15% ▪ ↓ LVEF: 35%	▪ No data	▪ 30 day mortality: 33.3%	Jain et al. [16]
164 (61) 78%	ICU	▪ RV Dys: 35% ▪ RV Dilat: 38% ▪ LV normal ▪ D-Dimer/TnI correlated with RV Dys	▪ No data	▪ 30 days mortality: 40% ▪ Highest mortality with ▪ RV Dys	Moody et al. [17]
1216 (62) 70%	ICU 60%	▪ LV Dys: 39% ▪ RV Dys: 33% ▪ Severe Dys or tamponade: 15% ▪ AMI: 3% ▪ Myocarditis: 3% ▪ Takotsubo: 2%	▪ No data	▪ ↑ TnI/BNP predicts LV ▪ and RV abnormalities	Dweck et al. [18]
74 (59) 78%	ICU	▪ RV Dilat: 41% ▪ RV Dys: 27% ▪ Correlated with D-Dimer-CRP, not with TnI ▪ LV Normal	▪ PE in 20% of pts with RV Dys	▪ Mortality: 38%, more frequent with RV Dilat/Dys	Mahmoud-Elsayed et al. [19]
510 (64) 66%	ICU 68%	▪ RV Dilat: 35% ▪ RV Dys: 8% ▪ LVEF: 45% vs. 54% if RV Dys	▪ sPAP higher with RV Dilat	▪ Mortality: 32% ▪ RV Dys/Dilat associated with mortality, especially with↑TnI and D-dimers	Kim et al. [20]
90 (52) 74.4%	ICU (ECMO 42%)	▪ RV radial Dys: 6% ▪ RV longitudinal ▪ Dys: 24%	▪ ↑ sPAP ▪ ↑ PVR ▪ ↓ RV/PA coupling: 86%	▪ Death not reported ▪ RV Dys correlated with TnI, BNP, PVR	Bleakley et al. [21]
24 (64.5) 54&	NA	▪ LV Dys: 37% ▪ RV Dys: 17% ▪ RV + LV Dys: 17% ▪ ↑ TnI 100% ▪ Pericardial effusions: 33%	▪ NA	▪ Not Reported	Sud et al. [22]
86 (58.8) 60%	ICU 37%	▪ LV diastolic Dys: 66% ▪ ↓ LVEF: 17.5% ▪ RV Dys: 18.5%	▪ ↑ sPAP in deteriorating pts	▪ Mortality: 12.8% ▪ sPAP and LV diastolic ▪ Dys associated with mortality	Sattarzadeh Badkoubeh et al. [23]
224 (69) 62%	ICU 33%	▪ RV Dys in pts with PE ▪ LV Normal	▪ ↑ sPAP: 14%	▪ Mortality: 30% ▪ PE independent predictor of mortality	Scudiero et al. [24]
94 (64) 74%	ICU	▪ TAPSE/sPAP in non survivors (RV/PA uncoupling)	▪ ↑ sPAP in non survivors	▪ Mortality: 26% ▪ TAPSE/sPAP predicts mortality (multivariate)	D’Alto et al. [25]
305 (63) 67%	ICU 44%	▪ ↑ TnI: 62% ▪ RV Dys: 26% ▪ LV Systolic Dys: 18% ▪ LV Diastolic Dys: 13%	▪ No data	▪ Normal TnI: 5.2% mortality ▪ ↑ TnI + echo abnormalities: 31% mortality	Giustino et al. [26]
28 (61.7) 79%	ICU (ECMO 14%)	▪ ↑ TnI: 39% ▪ ↓ LVEF: 21% ▪ Acute cor pulmonale in 2 pts (7%)	▪ ↑ sPAP in all pts at ICU admission	▪ ↑ TnI correlated with CRP levels, but not with echo findings. ▪ Mortality 7% (but study ongoing on time of publication)	Lazzeri et al. [27]
67 (61) 82%	ICU	▪ ↑ TnI: 72% ▪ Hierarchical clustering method identified 4 clusters of pts according to cardiac preload, LV afterload, LV and RV contractility	▪ Pulmonary circulatory dysfunction ▪ graded as moderate (12%) or severe (46%)	▪ Mortality 39% ▪ ↑ TnI inversely correlated with indices of LV and RV contractility ▪ ↑ TnI and LV diastolic dysfunction correlated with mortality	Bagate et al. [28]

Abbreviations: AMI: acute myocardial infarction. ARDS: acute respiratory distress syndrome. BNP: brain natriuretic peptide. CRP: C-reactive protein. ICU: intensive care unit. LV: left ventricle. LVEF: LV ejection fraction. MV: mechanical ventilation. PA: pulmonary artery. PE: pulmonary embolism. PH: pulmonary hypertension. Pts: patients. PVR: pulmonary vascular resistance. RV: right ventricle. RV Dilat: RV dilation. RV Dys: RV systolic dysfunction. S′: tissue doppler-derived tricuspid lateral annular systolic velocity. sPAP; systolic pulmonary artery pressure. TAPSE: tricuspid annular plane systolic excursion. TnI: Troponin I.

**Table 3 jcm-10-02535-t003:** Main effects and side effects of vasopressors and inotropes for the treatment of RV dysfunction.

Drug	SVR	PVR	PVR/SVR	Main Adverse Effects
Vasopressors				
Norepinephrine	↑↑↑	↑	→/↓	↑ PA pressure (at >0.5 mg/kg/min), tachycardia
Phenylephrine	↑↑	↑↑	→	↑ PA pressure, ↑ RV afterload
Vasopressin	↑↑↑	→/↓	↓↓	Digital and mesenteric ischemia (keep < 0.03 U/min)
Inotropes				
Dobutamine	→/↓	→/↓	→/↓	Tachycardia, ↑ myocardial O_2_ demand, hypotension
Epinephrine	↑↑↑	↑↑	→/↓	Tachycardia, ↑ myocardial O_2_ demand, ↑ RV afterload
Milrinone	↓↓	↓↓↓	↓↓	Hypotension, tachycardia, ↑ myocardial O_2_ demand
Levosimendan	↓↓↓	↓↓	→/↓	Hypotension

Abbreviations: PA: pulmonary artery; PVR: pulmonary vascular resistance; SVR: systemic vascular resistance; RV: right ventricle.

**Table 4 jcm-10-02535-t004:** Summary of studies investigating inhaled nitric oxide in COVID-19.

Reference	Design	*n*	Population	[iNO]	iNO Duration	Effect on P/F O_2_	Effect on RV	Effect on PVR
Abou-Arab, O. et al. [151]	Prospective	34	ICU	10 ppm	30 min	Significant (20% increase in 65% pts)	Similar incidence of ACP in responders and non-responders	NR
Tavazzi, G. et al. [150]	Retrospective	16	ICU	25 ppm	30 min	Not significant (20% increase in 25% pts)	Better improvement of P/F O_2_ in pts with RV dysfunction	NR
Longobardo, A. et al. [152]	Retrospective case–control	27	ICU	10–20 ppm	NR	Not significant (10% increase in 40% pts)	NR	NR
Safaee Fakhr, B et al. [153]	Prospective observational	6	Obstetric/ICU	200 ppm (SB); 40 ppm (MV)	30 min 2×/day (SB) Continuous administration (MV)	Significant increase after each inhalation period	NR	NR
Ferrari, M. et al. [154]	Retrospective	10	ICU	20 ppm	30 min	Not significant	NR	NR
Garfield, B. et al. [155]	Observational	36	ICU	20 ppm	24 h (144 h median)	Significant (30% increase in 57% pts)	NR	NR
Lotz, C. et al. [149]	Retrospective observational	19	ICU	20 ppm	NR	Significant (20% mean increase)	NR	Median decrease of 15.9% (not significant)
Roba, c. et al. [156]	Prospective	9	ICU	20 ppm	1 h	Significant increase of P/F O_2_ and of cerebral saturation	NR	NR

Abbreviations. ICU: intensive care unit; iNO: inhaled NO; MV: mechanical ventilation; NR: not reported; RV: right ventricle; ppm: parts per million; SB: spontaneous breathing; [ ]: concentration.

## Data Availability

No data availability statement required for this article (REVIEW ARTICLE, no original data produced).

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
