# Peer review of "The Right Ventricle in COVID-19"

_jcm, 2021, doi:10.3390/jcm10122535_

Round 1
Reviewer 1 Report
The present manuscript addresses a hot topic in clinical practice, that is the clinical role of RV dysfunction in COVID disease, and its treatment. This is a comprehensive, well written review. However some issues can be raised.
a) A clear definition of RV dysfunction should be stated. RV dilatation and dysfunction may not be the same clinical (and echocardiographic) entity, since RV dilatation is not always associated with dysfunction and treatment may be different. Evidence should be clearly described and discussed.
b) Tables summarizing reports (including study population characteristics, main results and conclusions/limitations) should be added to the present version of the manuscript. We suggest to distinguish between ICU and non ICU patients, since RV alterations seem to be related to disease severity, as well as to mechanical ventilation.
c) Table 1: we suggest a clearer definition of study population. ICU and non ICU should be preferred. The term MV (and percentages) is not easy to understand. Does it mean “critical and not critical patients, with different percentages of mechanically ventilated patients”?
d) We suggest to add a paragraph describing technical difficulties frequently encountered in COVID patients, when performing an echocardiogram (in particular with techniques such as speckle tracking) and the need of protection of staff.
e) Some considerations about other comorbidities should be added, since they often coexist and affect dhe course of the illness and the prognosis, e.g. hypertension, diabetes, coronary disease (see Guo T et al, 2019 and Shi S et al, 2020) and overweight too. Consequently, ECG and Echo alterations could exist before COVID-19 and therefore they do not relate to the infection (see Deng Q et al, 2020).
f) Are RV alterations related to disease severity? This concept should be better assessed and discussed.
fi) The pathophysiologic mechanisms of RV alterations in COVID disease are well described.
fii) Evidence on potential pharmacologic treatment do not seem complete. For instance , why did the Author choose to describe and discuss only some and not all papers on iNO treatment in COVID disease?
fiii) Data on VV ECMO and COVID are quite heterogeneous. We suggest the Authors to perform a systematic and complete revision of evidence on this topic or to delete the paragraph entitled “ECMO”. We believe that reporting a mortality rate of 15% in ECMO COVID patients does not reflect the worldwide scenario nor the growing evidence. The clinical and pathophysiologic role of ECMO in COVID disease is far to be completely understood.
fiv) References should updated: Dandel M Infection 2021, Dandel M Heart Fail Rev 2021, Jain R et al J Patient Cent Res Rev. , Lazzeri C et al Am J Cardiol, Bagate F, J Intensive Care 2021
Reviewer 2 Report
Dear Editors, Dear Authors
Thank you for giving me an opportunity to review this manuscript. It is well written, with adequate flow and broad description of the subject. This article is a comprehensive review of the relevant literature pertaining to right ventricle dysfunction apparently present in up to 40% of patients with COVID-19.
Minor comments:
Line 33: is “taksotsubo”, should be “takotsubo”
Line 56: the authors write: “ventilator management in the ICU.” In my opinion it should read “respiratory failure management in the ICU”, which is a more complex process than just initiating mechanical ventilation.
Table 1 should include a column (usually the first one) to present the last name of the first author, e.g. Ferrante et al. [3].
Line 471 - Mechanical ventilation - please add available information on the effect of Prone position and ventilation in Prone position on RV strain.
The manuscript has only 2 Figures. Please be more “graphic” and try to illustrate the pathophysiology described in details in form of a figure.
Moreover, “Specific management of RV failure” also deserves a decent figure.
A table summarising the effect of different vasopressors and inotropes, along with their side effects would substantial to help clinicians deal with RV strain or failure and make educated clinical decisions.
I would like to congratulate the Authors on the choice of subject and high quality of their review. However, I deeply believe that all of the above-mentioned suggestions would make the manuscript even better and more valuable for the bedside clinician.
With best regards
Round 2
Reviewer 1 Report
The clinical role of RV dysfunction in COVID disease, and its treatment is really a hot topic in clinical practice. I commend the Authors for accurately, completely and quickly responding to the issues raised. I therefore recommend publication of the review.